# Redefining Neurodevelopmental Impairment: Perspectives of Very Preterm Birth Stakeholders

**DOI:** 10.3390/children10050880

**Published:** 2023-05-14

**Authors:** Anne Synnes, Amarpreet Chera, Lindsay L. Richter, Jeffrey N. Bone, Claude Julie Bourque, Sofia Zhang-Jiang, Rebecca Pearce, Annie Janvier, Thuy Mai Luu

**Affiliations:** 1Department of Pediatrics, University of British Columbia and BC Children’s Hospital Research Institute, Vancouver, BC V5Z 4H4, Canada; 2Department of Pediatrics, University of British Columbia, Vancouver, BC V6T 1Z4, Canada; 19akb6@queensu.ca (A.C.); lrichter@bcchr.ca (L.L.R.); sofiazj@student.ubc.ca (S.Z.-J.); 3BC Children’s Hospital Research Institute, Vancouver, BC V5Z 4H4, Canada; jbone@bcchr.ca; 4CHU Sainte-Justine, Montreal, QC H3T 1C5, Canada; claude.julie.bourque@umontreal.ca; 5Science and Mathematics Education Research Group, Faculty of Education, McGill University, Montreal, QC H3A 0G4, Canada; rebecca.pearce@gmail.com; 6Department of Pediatrics, Bureau de l’Éthique Clinique (BEC), Université de Montréal, QC H3C 3J7, Canada; anniejanvier@hotmail.com; 7Division of Neonatology, Unité d’Éthique Clinique, Unité de Soins Palliatifs, Bureau du Partenariat Patients-Familles-Soignants, Montréal, QC H3T 1C5, Canada; 8Department of Pediatrics, Université de Montréal, CHU Sainte-Justine, Montréal, QC H3T 1C5, Canada; thuy.mai.luu@umontreal.ca

**Keywords:** prematurity, outcomes, patient-oriented research, neurodevelopment, stakeholders

## Abstract

Children born very preterm are at risk of severe neurodevelopmental impairment, a composite endpoint that includes cerebral palsy, developmental delay, and hearing and visual impairment defined by medical professionals. We aimed to describe preterm birth stakeholders’ perspectives on this classification. Ten clinical scenarios describing 18-month-old children with different components of severe neurodevelopmental impairment and one scenario of a typically developing child (control) were distributed to parents and stakeholders using a snowball sampling technique. For each scenario, participants rated health on a scale from 0 to 10 and whether the scenario represented a severe condition. Results were analyzed descriptively and mean differences from the control scenario were compared using a linear mixed-effects model. Stakeholders (number = 827) completed 4553 scenarios. Median health scores for each scenario varied from 6 to 10. The rating for the cerebral palsy and language delay scenario was significantly lower (mean difference −4.3; 95% confidence interval: −4.4, −4.1) than the control. The proportion of respondents rating a scenario as “severe” ranged from 5% for cognitive delay to 55% for cerebral palsy and language delay. Most participants disagreed with the rating used in research to describe severe neurodevelopmental impairment in preterm children. The term should be redefined to align with stakeholder perceptions.

## 1. Introduction

Prematurity, defined as a live birth before 37 completed weeks’ gestation, is a global problem affecting more than 10% of births and 15 million babies every year [1]. Preterm birth rates vary. In 2021, of all live-born children in the USA, 10.5% were born preterm [2] and in Canada, 8% were born preterm with 6.4% born at 33–36 weeks’, 1% at 29–32 weeks’, and 0.6% born extremely preterm at <29 weeks’ gestation [3]. 

Internationally, prematurity is responsible not only for 35% of newborn deaths and 18% of deaths before age 5 years [1] but is also associated with an increased risk of neonatal complications, long-term morbidities [4], and higher healthcare costs [5,6] with the highest impact in the extreme preterm population. Evaluating outcomes of prematurity is essential for clinicians, parents, and the healthcare system to improve the future of children born preterm.

Unlike survival, the evaluation of long-term outcomes is challenging. In the last 50 years there have been numerous studies documenting long-term neurodevelopmental and health outcomes associated with prematurity. The use of standardized developmental tests to evaluate preterm outcomes objectively started with Drillien’s study of physical growth and mental development in children born weighing less than 3 pounds in Edinburgh between 1953 and 1955 [7]. The use of standardized neurodevelopmental tests became a common outcome measure in the scientific community thereafter. Cerebral palsy and visual and hearing impairments occur more frequently in the preterm population [4] and it became common practice to report a composite outcome of severe neurodevelopmental impairment which included cerebral palsy, deafness, blindness, and a score of below −2 or −3 standard deviations below the mean on a standardized neurodevelopmental test [4,8]. To minimize attrition, a neurodevelopmental assessment will often occur at a corrected age of 18–36 months. At this age the Bayley Scales of Infant and Toddler Development [9] is a frequently used developmental test. Neonatal follow-up programs incorporate these assessments into standard care [10,11]. The reported research using these outcomes is the foundation for life and death decisions such as resuscitation at the threshold of viability [12,13].

Communication with parents, as legal guardians and advocates for their child, is essential and should be parent-personalized [14]. Healthcare professionals need to provide parents with the information they need in language they understand. The available research data frequently use the term “severe neurodevelopmental impairment”, a composite outcome that includes several sensory and neurodevelopmental components as the primary outcome in observational studies [15,16,17] and clinical trials [8]. Composite outcomes require that the individual components are of equal significance [18,19,20]. Yet, there is a paucity of research on whether the individual components have a similar impact on the child and family. Preterm outcome research has been dominated by the use of researcher-determined outcomes and definitions and parents have not been asked what outcomes are important to them. 

The goal of our research, part of the patient-oriented research network Child Health Initiatives Limiting Disability - Brain Research Improving Growth and Health Trajectories [21] was to identify outcome measures that are meaningful to parents of children born very preterm. In this study, our aim was to investigate stakeholders’ perspectives on the classification and definitions of neurodevelopment used to describe children born preterm. Stakeholder perceptions of the Canadian Neonatal Follow-Up Network (CNFUN) [22] definition of “severe” neurodevelopmental impairment were explored, as well as the equivalence of the components used to define severe neurodevelopmental impairment.

## 2. Materials and Methods

The opinions of preterm birth stakeholders including parents, individuals born preterm, healthcare professionals, researchers, trainees, and educators were captured using a cross-sectional survey. 

### 2.1. Development of the Clinical Scenarios

The composite outcome, severe neurodevelopmental impairment, used by the CNFUN has similar components to those outcomes used by other neonatal outcome researchers. Information is collected at an in-person assessment at a targeted age of 18–21 months corrected age. Severe neurodevelopmental impairment includes one or more of the following: Bayley Scales of Infant and Toddler Development third edition (Bayley-III) [9] cognitive, motor, language, or general adaptive cognitive score < 70 (less than 2 standard deviations below the mean), need for hearing aids or cochlear implants, bilateral visual impairment (one or more of the following: no response to a 1 cm object, small eye, corneal scarring, sustained sensory nystagmus, ophthalmologist report of retinopathy of prematurity stage 3 or greater, or a report of visual acuity of 20/70 or worse), or cerebral palsy with a Gross Motor Function Classification System [23] level 3 or higher. 

Ten easy to understand clinical scenarios were created to align with the CNFUN definition of severe neurodevelopmental impairment (Appendix A). Six scenarios captured children with a single “severe” impairment (cerebral palsy, Bayley-III motor, language, or cognitive score < 70, hearing or visual impairment). Five scenarios represented the most common combinations of impairments observed in the CNFUN database of very preterm infants. Finally, an eleventh scenario, used as a control condition, described a typically developing child. The scenarios were developed, reviewed, and edited for face validity by an interdisciplinary team of neonatal follow-up healthcare providers at British Columbia’s Women’s Hospital Neonatal Follow-Up Program and for acceptability by parent representatives. For each scenario, respondents were asked whether the fictional child did or did not have a severe health condition and to rate the severity of the health states on a scale from 0 (worst possible health) to 10 (best possible health). Scenarios were translated from English to French and back-translated to English for validity. Parent reviewers recommended reducing the questionnaire burden by limiting each questionnaire to six scenarios. Twelve questionnaires with different combinations of six scenarios, all including the scenario of the control child, were created using the Research Electronic Data Capture platform. Participants were randomized to receive one of the twelve questionnaires. 

### 2.2. Survey Administration

Eligible participants were recruited in two steps: a local pilot study followed by a wide distribution of the questionnaire. During the pilot step, parents of children born at less than 29 weeks’ gestation attending a clinic visit at 3 to 5 years of age at the British Columbia’s Women’s Hospital Neonatal Follow-Up Program in Vancouver, Canada, were invited to participate. During the second step, an online snowball-sampling method was used [24] and respondent eligibility expanded to Canadian and international stakeholders. Demographic questions were added to describe the various stakeholder groups. The initial link to the online survey originated within the private Facebook groups of the parent stakeholder partners Canadian Premature Babies Foundation, Préma-Québec, and the Child Health Initiatives Limiting Disability—Brain Research Improving Growth and Health Trajectories network. Respondents were encouraged to share the link with others and the invitation was distributed to online parent-support sites. The survey was available in English or French from April 2021 to June 2021.

### 2.3. Statistical Analysis

Respondents were described and compared for each of the sampling populations (pilot phase, national, and international snowball samples) to evaluate whether the population survey results could be combined. 

The scenarios were ranked according to median and mean health-scores. In our primary analysis we treated the health scores as continuous data and estimated adjusted mean differences between the control and severe neurodevelopmental impairment scenarios with 95% confidence intervals using a linear mixed-effects model. We adjusted for potential differential responses between the pilot phase and the snowball sample and Canadian and international respondents, and by respondent characteristic (parent versus other) by including interaction terms. As a sensitivity analysis, the responses were treated ordinally, and a mixed-effects proportional-odds model was fitted. In all models, random intercepts were included. 

The proportion of scenario responses that were perceived as a severe health condition were calculated for each scenario. A mixed-effects logistic regression model, adjusting for the same confounders as above was used, with a binary response for each scenario of “severe” vs. “not severe”. Results were summarized as adjusted odd ratios and 95% confidence intervals. 

### 2.4. Ethics

Participants were informed about the goal of the study and informed consent was sought at the beginning of the survey. The survey was anonymous and only accessible if consent was given. Research ethics board approval was obtained from the University of British Columbia Children’s and Women’s Research Ethics Board (H17-03490).

## 3. Results

### 3.1. Population Characteristics

Overall, 827 participants responded to a total of 4553 clinical scenarios. During the pilot phase, 62 parents attending a neonatal follow-up clinic in Vancouver completed the questionnaire. In the snowball step, 765 stakeholders answered the questionnaire (442 Canadian and 323 international). Participant demographics were obtained from respondents in the snowball step. Respondent age varied from less than 20 years to a maximum of more than 65 years with 47% of the participants aged 31 to 40 years old. Surveys were completed in English by 43% and in French by 57% of respondents. Most (60%) participants were parents, children, or family members and 28% were healthcare professionals (Table 1). 

The regression analyses, using mixed-effects models to evaluate the effect of the recruitment sample, showed that the responses of parents in the pilot study and stakeholders in the second stage of the study were similar with no statistical association with scenario rating. (Figure 1) Results were therefore combined for subsequent analyses. The language of the questionnaire, English versus French, was also not associated with responses. 

### 3.2. Severity Rating

The median, mean, and distribution of severity ratings of the clinical scenarios are shown in Figure 2. As expected, the typically developing child was perceived to represent the best possible health state with a median score of 10. The other scenarios had median scores between 6 and 8 with considerable spread. 

Figure 3 shows that the mean differences and 95% confidence intervals between the control and other scenarios were all statistically significantly different. The scenario which describes a child with cerebral palsy and language delay was perceived to be the lowest-rated health condition with a mean difference from the control of −4.28 (95% confidence interval −4.43, −4.13) whereas the scenario describing a child with significant cognitive delay showed the smallest mean difference of −2.04 (95% confidence interval −2.21, −1.88). Results were similar when the responses were treated ordinally. 

### 3.3. Classification of Scenario as a Severe Health Condition

Figure 4 displays the percentage of respondents who perceived the impairment of the child described in the scenario as being severe. For 8 of the 10 scenarios representing a “severe neurodevelopmental impairment”, fewer than 50% of the respondents perceived the health condition as severe. The scenario representing a child with cerebral palsy and language delay was considered severe by the largest percentage (55.5%) and a visual impairment rated severe by 51% of respondents. The scenario representing a child with cognitive delay was classified as severe by the smallest proportion (4.7%) of respondents. 

### 3.4. Comparison of Respondent Characteristics

There were no statistically significant differences in severity rating between respondents recruited during the pilot phase in the neonatal follow-up clinic and in the national and international surveys. Non-parent (healthcare professionals, children born preterm, teachers, researchers, trainees) answers were similar to parents, except for the cerebral palsy and language delay scenario, where non-parents rated the scenario slightly lower than parents with a mean difference of −0.36 (95% confidence interval −0.66, −0.05) (Table 2). Respondent gender, age, and language of survey completion were not associated with survey responses. 

## 4. Discussion

In this study we captured stakeholders’ perspectives regarding the rating and definition of severe neurodevelopmental impairment following very preterm birth. The 10 different health conditions that represent traditional severe neurodevelopmental impairments had median health scores ranging from 6 to 8 on a scale from 0 to 10. As expected, the health scores were all significantly lower than the typically developing child scenario. Importantly, most respondents did not perceive the scenarios to represent severe health conditions. Yet, all scenarios illustrated a case in which the child would have been labelled as having a “severe” or “significant” impairment according to definitions used in many neonatal outcome studies. 

Our first finding is that stakeholders, parents, and clinicians generally rated the clinical scenarios more favorably than expected. This is important because the term “severe neurodevelopmental impairment” is used to make professional recommendations about life and death decisions. These outcomes are also used to communicate with parents and prepare them for the future. Our results identify the potential for miscommunication when the term “severe” is used. Indeed, severe can have a different meaning for different people. For some, it may indicate that a condition is more serious and therefore requires more attention. For others, the word may evoke more negative qualifiers such as undesirable or bad, which may promote stigma that can particularly affect individuals with developmental disabilities and their family [25]. This could be potentially avoided by describing outcomes using neutral descriptions such as grading and staging, as used for certain medical conditions. For example, the term “severe hearing impairment” can be replaced with “the need for hearing aids or a cochlear implant to function” which describes the practical impact of the condition without a judgement on the desirability of the condition. There is a need to review the terminology used to describe outcomes after preterm birth by including stakeholders’ perspectives.

Our second finding is that not all impairments classified as severe by clinicians and researchers are comparable: is being blind worse than being deaf? In our study a visual impairment was considered worse than hearing loss. Yet, these individual conditions are often combined into a composite endpoint in neonatal outcome studies and trials. Composite outcomes are useful to increase trial statistical efficiency, but rules exist to ensure their validity and interpretability [18,19,20]. One key element is that individual components should carry similar importance to stakeholders. Our results show that the definition of severe neurodevelopmental impairment used in this study does not meet the criteria of similar importance. Furthermore, for many scenarios, there was a wide variability in the responses indicating the subjective nature of the term “severe”. Cerebral palsy may be considered worse by parents who are professional athletes, compared to parents who live with a motor disability, and hearing loss may be considered more impactful by parents who are musicians. Our findings question whether using the current CNFUN definition of severe neurodevelopmental impairment is valid. The same conclusion is likely to be generalizable to other similar composite outcomes.

Our third finding is that, overall, clinicians and parents/families who participated in this study had similar responses. This is in contrast with studies showing that quality of life is valued differently by healthcare providers, parents, and adolescents born very preterm: clinicians generally being more pessimistic than parents and preterm survivors [26]. Webbe et al. also reported that the importance of neonatal outcomes was considered differently by clinicians and parents [27]. Other studies using clinical vignettes in neonatology have demonstrated a similar phenomenon: when study participants are exposed to a story with descriptive outcomes or functional outcomes (as opposed to terms like “a 24-weeker” or “severe disability”) clinicians tend to value patients more and/or non-clinicians and clinicians tend to answer similarly [28]. However, severe is a subjective term and we identified variability amongst our respondents. 

Our results should be interpreted with consideration of the clinical scenarios used in the study. The scenarios described the developmental status of young children and not their future trajectories. This was necessary, especially for the clinical scenarios representing Bayley-III scores since there are limitations to their predictive ability [29]. For example, about half the children with “severe neurodevelopmental impairment” were not categorized as “severe” when evaluated at 5 years of age in the Caffeine for Apnea in Prematurity trial [8]. Severe neurodevelopmental impairment encompasses a spectrum of health conditions. The scenarios used in this study described less severe conditions on this spectrum. 

Our mixed-methods research program aimed to capture the voices of parents about their children who were born preterm and address biases and barriers to participation by using qualitative [30] and quantitative [31,32] methods, including parents with participation barriers [33] and, in this study, a broader range of stakeholders using scenarios. In these studies, parents reported that positive, as well as negative, aspects of very preterm children’s health and development [30,31] should be reported and that there are other outcomes such as nutrition, rehospitalizations, behavior, and sleep that are important to them [31,32].

In neonatal follow-up research [4,8,15,16,17,34] most outcomes have been selected by individual groups of neonatologists, researchers, or clinicians without consideration for what parents or families consider important or impactful. Parents need appropriate information described in understandable terms to make the best decision for their children. Considering our results, more consideration is warranted by researchers and clinicians in the choice and definition of outcome measures and how information is shared with families. We recommend avoidance of the term “severe” and propose that objective practical descriptors, such as “need for hearing aids or cochlear implant to function” should be used. The terminology to describe the functional impact of outcomes needs to be developed together with parent and stakeholder partners. Our study provided a voice for parents of children born preterm and other key stakeholders to guide how outcomes should be reported during follow-up and research programs. 

This study had some limitations. Several factors may have affected responses to these hypothetical scenarios. Respondent characteristics and lived experiences vary and may change over time. Parents who are active on online support groups are possibly different from parents who are not; the same can be said about clinicians. This community of stakeholders is likely to influence new parents of preterm infants who turn to online support groups. These are also self-reported questionnaires with their inherent biases, limiting the participation of parents with limited literacy or who speak different languages. On the other hand, parent respondents in the local pilot group, who were not affected by these sampling biases, had similar answers. The large number of respondents from different backgrounds ensured that many diverse stakeholder voices were heard. 

## 5. Conclusions

This study shows that the assessment of severity based on a clinical scenario differs from commonly used definitions of severe neurodevelopmental impairment used in outcomes research. Secondly, it highlights the problems of using a composite outcome to describe neurodevelopmental impairment. Optimizing the way neonatal outcomes are reported is essential. Further research is needed to identify which outcomes are most meaningful to parents and how to effectively communicate outcomes to families. 

## Figures and Tables

**Figure 1 children-10-00880-f001:**
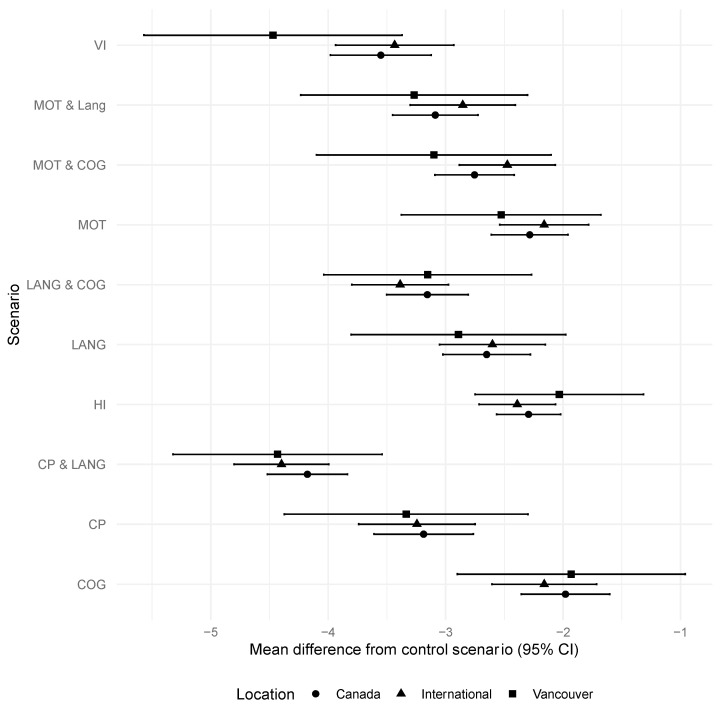
Mean difference between scenario and control by population samples. CI, confidence interval; CP, cerebral palsy; MOT, motor impairment; COG, cognitive impairment; LANG, language delay; VI, visual impairment; HI, hearing impairment.

**Figure 2 children-10-00880-f002:**
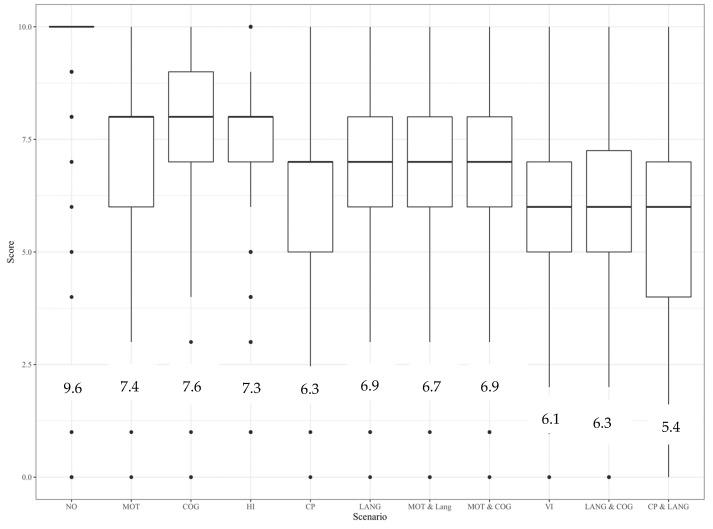
Boxplot of distribution of ratings of clinical scenarios. Median values are shown in the whisker plots and means appear numerically underneath. NO, no impairment; CP, cerebral palsy; MOT, motor impairment; COG, cognitive impairment; LANG, language delay; VI, visual impairment; HI, hearing impairment.

**Figure 3 children-10-00880-f003:**
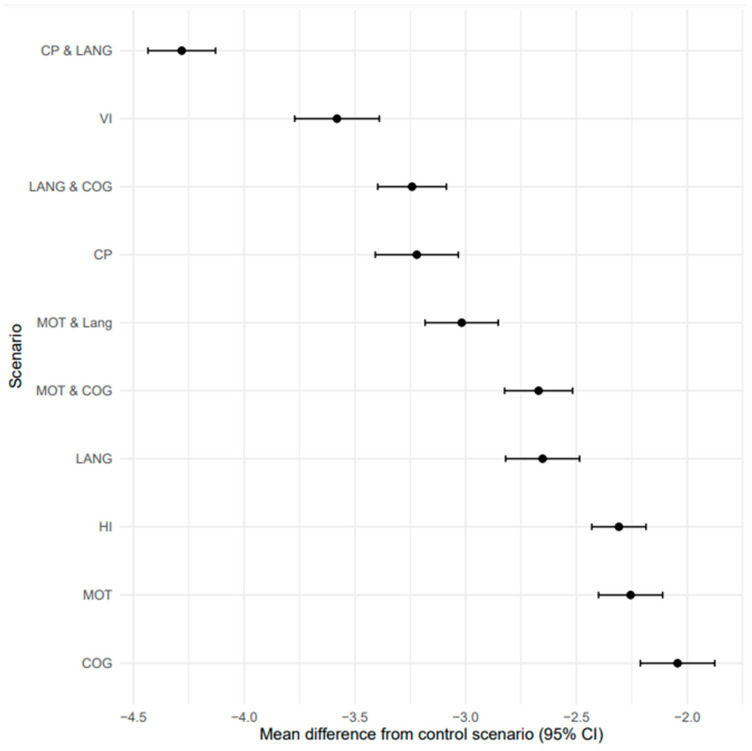
Mean differences in ratings between scenarios representing a severe neurodevelopmental impairment condition and the scenario describing a typically developing child. Results show mean differences and 95% confidence intervals (CIs). NO, no impairment; CP, cerebral palsy; MOT, motor impairment; COG, cognitive impairment; LANG, language delay; VI, visual impairment; HI, hearing impairment.

**Figure 4 children-10-00880-f004:**
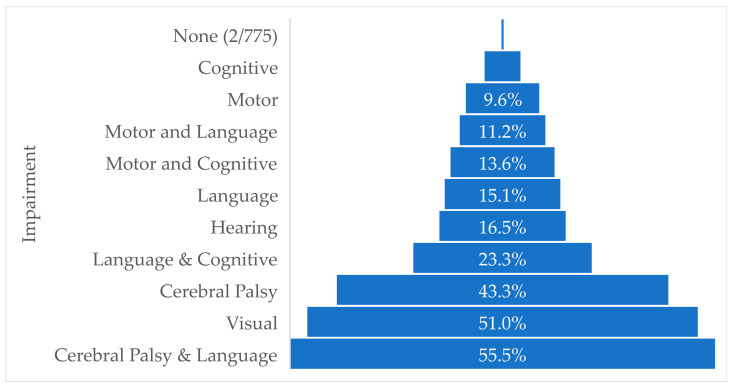
Proportion of clinical scenarios perceived to be a severe health condition. Results exclude missing and ‘don’t know’ responses.

**Table 1 children-10-00880-t001:** Characteristics of participants who completed the survey by the snowball method.

	Characteristic	N (%) (N = 827)
Language of Questionnaire	English	354 (43%)
	French	473 (57%)
Personal or Professional Description	Parent/caregiver or family of a child born preterm	471 (57%)
	Person born preterm	21 (3%)
	Healthcare professional	228 (28%)
	Teacher/educator	14 (2%)
	Student/trainee	5 (1%)
	Researcher	14 (2%)
	Other	9 (1%)
Country of Residence (international survey only)	United States	54 (6.5%)
	United Kingdom	26 (3.1%)
	France	181 (21.9%)
	Other	62 (7.5%)

N = number.

**Table 2 children-10-00880-t002:** Comparison of parent and non-parent respondents for severity ranking of different scenarios.

Scenario	Mean Differences (95% CI)	*p* Value
No impairment	−0.06 (−0.3, 0.17)	0.60
Cerebral palsy	−0.23 (−0.61, 0.16)	0.25
Motor	−0.04 (−0.32, 0.25)	0.81
Cognitive	0.25 (−0.09, 0.59)	0.16
Language	0.14 (−0.2, 0.48)	0.41
Visual impairment	0.03 (−0.36, 0.41)	0.89
Hearing impairment	−0.22 (−0.46, 0.2)	0.07
Motor and language	0 (−0.34, 0.33)	0.99
Language and cognitive	0.24 (−0.08, 0.55)	0.14
Motor and cognitive	−0.03 (−0.34, 0.28)	0.83
Cerebral palsy and language	−0.36 (−0.66, −0.05)	0.02

CI, confidence interval.

## Data Availability

The data presented in this study are available on request from the corresponding author.

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
