# Peer review of "Redefining Neurodevelopmental Impairment: Perspectives of Very Preterm Birth Stakeholders"

_children, 2023, doi:10.3390/children10050880_

Round 1

Reviewer 1 Report

Dear colleagues,

Thank you very much for submitting your manuscript.

The study of neuro-motor development problems in children born prematurely represents a problem with significant implications for medical practice.

The text is clear and easy to read. The manuscript has an excellent structure and description. The overall paper is organized and well-written. The literature reviews are insightful and informative.

The figures and the tables are well-presented and easy to read and understand. The presented aspects sufficiently support the conclusions.

I have only a few remarks to make:

The Abstract section has no conclusion.

You should enter a few more keywords considering the topic of broad interest addressed.

Regarding the article's purpose, several aspects are described at the end of the Introduction section and the beginning of the Material and Method section. It would be more useful if the purpose of the work were inserted only at the end of the first section.

Figure 4 needs to be included.

The study's limitations are recommended at the end of the Discussions.

I congratulate all the authors for their efforts.

Author Response

Thank you for your helpful comments which are numbered with responses in italics.

  1. The Abstract section has no conclusion.

We have added the conclusion “The term severe neurodevelopmental impairment should be redefined to align with stakeholder perceptions. “ Though this has increased the abstract to 207 words we interpret that this still meets the guidelines for maximum words of “ of about 200 words maximum”

  1. You should enter a few more keywords considering the topic of broad interest addressed.

The additional key words neurodevelopment; stakeholders” have been added.

  1. Regarding the article's purpose, several aspects are described at the end of the Introduction section and the beginning of the Material and Method section. It would be more useful if the purpose of the work were inserted only at the end of the first section.

The first sentence under Material and Methods was amalgamated into the end of the Introduction “In this study, our aim was to investigate stakeholders’ perspectives on the classification and definitions of neurodevelopment used to describe children born preterm.   Stakeholder perceptions of the Canadian Neonatal Follow-Up Network (CNFUN) [22] definition of “severe” neurodevelopmental impairment were explored as well as the equivalence of the components used to define severe neurodevelopmental impairment.”

We felt it was appropriate to keep a description of the type of methodology as “a cross-sectional survey” and identify the key stakeholders under Materials and Methods.

  1. Figure 4 needs to be included.

Figure 4 has been added.

  1. The study's limitations are recommended at the end of the Discussions.

The paragraph on study limitations has been moved to the end of the Discussion.

Reviewer 2 Report

Thank you for the opportunity to review this article

Line 29 number should be written in full if starting the sentence or change the sentence around.

State earlier what the control was

Abstract needs concluding, maybe a recommendation

Line 63 - not clear what 'this' is referring to

Line 66 ? standards of care or standard care

Line 90 should this be clarified as to which category of preterm or is this for all categories

Line 188 incomplete sentence

figure 4 missing

Line 227 sentence not clear and too long

consistency with use of single or double quotation marks

Line 294 - need to say how this is a limitation

Wish well with this worthy research

minor issues only

Author Response

Thank you for your helpful comments.

  1. Line 29 number should be written in full if starting the sentence or change the sentence around.

The sentence has been changed to “Stakeholders (n=827)” to avoid starting the sentence with a number. Hopefully the inclusion of the commonly used abbreviation “n=” is acceptable as we are trying to stay within the maximum allowed word count..

  1. State earlier what the control was.

The (control) has been moved from line 29 to 26. This provided us with the opportunity to reduce the number of words in the abstract by replacing "typically developing child” with “control” in line 29.

  1. Abstract needs concluding, maybe a recommendation

We have added the conclusion and recommendation “The term severe neurodevelopmental impairment should be redefined to align with stakeholder perceptions. “Though this increased the abstract to 205 words we interpret that this still meets the guidelines for maximum words of “ of about 200 words maximum”

  1. Line 63 - not clear what 'this' is referring to

This has been clarified by replacing “this” with “a neurodevelopmental assessment”.

  1. Line 66 ? standards of care or standard care

Standards of care” has been changed to “standard care”.

  1. Line 90 should this be clarified as to which category of preterm or is this for all categories

Preterm birth stakeholders” refers to all preterm birth categories as defined in the first paragraph of the introduction. In section 2.2 the recruitment processes are described in detail. In the pilot phase participants were parents of children born at less than 29 weeks’ gestation and in the second online phase participants self-identified which preterm birth stakeholder group best described them. We have therefore elected not to change the wording in line 90.

  1. Line 188 incomplete sentence

Line 188 is a footnote for Figure 2 which includes the abbreviations used in the figure. Using the template this was inserted into the text and not easily differentiated from the text. We have moved the footnote adjacent to the figure title and added an extra line break to make this clear. We welcome editorial staff suggestions.

  1. figure 4 missing

Figure 4 has been added.

  1. Line 227 sentence not clear and too long

The sentence starting on line 227 has been divided into 2 sentences and the meaning should be clearer: “The 10 different health conditions that represent traditional severe neurodevelopmental impairments had median health scores ranging from 6 to 8 on a scale from 0 to 10. As expected, the health scores were all significantly lower than the typically developing child scenario.”

  1. consistency with use of single or double quotation marks

The manuscript has been reviewed. All quotation marks are now double.

  1. Line 294 - need to say how this is a limitation

Line 294 (line 316 in the revised manuscript) is not a description of a limitation but rather how the limitation has been mitigated.

In response to Reviewer 1’s suggestion to move the limitations to the end of the Discussion section, the last sentence has been revised to be a concluding statement: “The large number of respondents from different backgrounds ensured that many diverse stakeholder voices were heard. “

Reviewer 3 Report

Redefining neurodevelopmental impairment: perspectives of 2 very preterm birth stakeholders.

In this study, the authors aimed to describe preterm birth stakeholders’ perspectives on the traditional classification of severe neurodevelopmental impairment defined by medical professionals, which is a composite endpoint that includes cerebral palsy, developmental delay, hearing and visual impairment. They distributed ten clinical scenarios describing 18-month-old children with different components of severe neurodevelopmental impairment and one scenario of a typically developing child to parents and stakeholders using a snowball sampling technique, to explore their evaluation of the severity of each particular condition. The proportion of respondents rating a scenario as severe ranged from 5% for cognitive delay to 55% for cerebral palsy and language delay. The main finding of the study was that most participants disagreed with the rating used in research to describe severe neurodevelopmental impairment in preterm children.

The manuscript is very well written and the results are very interesting.

In my copy, Figure 4 is missing. Otherwise, there are no other aspects that should be modified. Excellent work.

Author Response

Thank you for your positive comments.

Figure 4 has been added to the revised manuscript.